# "Sucha Góra" ("Trockenberg")—The Triangulation Point Doomed to Be Forgotten?

Marzena Lamparska [1,*] and Mirosław Danch [2]

1 Assoc. Institute of Social and Economic Geography and Spatial Management, Faculty of Natural Sciences, University of Silesia in Katowice, ul. Będzińska 60, 41-200 Sosnowiec, Poland
2 Polish Society of Amateur Astronomers, al. Planetarium r, 41-500 Chorzów, Poland; mirek@danch.pl
* Correspondence: marzena.lamparska@us.edu.pl

**Abstract:** The current study documents the importance of research on preserved artifacts which were previously used to take measurements of the Earth, and their importance for cultural heritage. The article reviewed the available source documents presenting the history of the astrogeodetic control point of Sucha Góra-Trockenberg as a monument of the first order triangulation network, preserved in cartographic materials and as the starting point of local geodetic networks, used in mining until 2000 in the so-called Upper Silesian Coal Basin, located in the territory of Poland and the Czech Republic. In order to determine the history of the triangulation work and the dates that the geographic coordinates of the peak were determined, field journals and other available materials were analyzed. The interesting location of this astrogeodetic control point, being in the vicinity of a UNESCO site, as well as its location in a forest and park complex, justify undertaking activities related to the conservation and promotion of this cultural heritage site.

**Keywords:** cultural heritage; triangulation points; Earth measurements

## 1. Introduction

The cultural heritage of geodesy, cartography, and geography is ambiguous [1]. It has an important scientific and humanistic aspect: the history of measuring the Earth's surface and the international cooperation of scientists representing many countries. The set of geodetic markers delineating 26°43′ E (the meridian is now on the UNESCO World Heritage List) is one of the great material monuments characterizing the heritage of geodesy as the cultural heritage of the World [2]. Unfortunately, the language of geodesy and geopolitics is also the language of colonial times, marking domains and dividing the world, as demonstrated by Capello, Rainsford and Aldrich [3–5]. The industrial development that began in Europe in the 18th century required detailed maps. More and more accurate measurements of the Earth became inevitable in the development of administration, economy, and industry, and for the implementation of military tasks. Between the 18th and 19th centuries, triangulation-based measurement systems began to be formed. Creating geodetic networks facilitated the designation of space for marking accurate coordinates. The precision of the geodetic systems created between the 18th and 20th centuries became the basic reference for all planning and engineering works, until modern times, when terrestrial measurement was replaced by aerial and satellite survey.

The intriguing history of the astrogeological checkpoint "Sucha Góra" (meaning "Dry Mountain" whose former German name was "Trockenberg"), the Prussian triangulation network that connected the Western European triangulation network, is presented here. Moreover, for over 150 years, this astrogeodetic control point acted as the origin point for all mine maps in the Upper Silesian Coal Basin in Poland, and in northern parts of the Czech Republic [6]. The artifact, which is the main subject of this article, is an inconspicuous granite pillar.

This point was one of 38 places in Europe that were selected as astronomical and geodesic measuring points in the Central European meridian project, which was later transformed into the project to measure the European meridian. The "Sucha Góra" triangulation point also belonged to the measurement line in the pan-European project to measure the length of arc of the 52° N parallel. In addition, it is a connection between three large measurement networks in Europe, created for economic and scientific purposes by the Kingdom of Prussia, the Russian Empire and the Austro-Hungarian Empire (it was the so-called Tarnowskie Góry connector). It was also the beginning of local and geodetic networks in Upper Silesia, on the basis of which measurements for the developing Upper Silesian mining industry were carried out. Systematized documenting of the facts as mentioned earlier has become the task of the authors.

The astronomy point "0" Sucha Góra is one of the most important points of the triangulation network developed in European measurements. It refers to the cultural and scientific heritage of historical astronomical observatories: the observatory in Greenwich, Pulkovo, Paris, or Ferro Astronomical [7]. Recalling its history, we document the resources of Sucha Góra's cultural heritage in terms of the history of Cartography, Geodesy, and Geography development.

Surveying points determine the processes of building Western civilization: they are also their witnesses and guardians: they document the emergence of new civilization spaces, they commemorate personalities related to philosophy and science, becoming landscape icons and monuments [8].

## 2. Materials and Methods

### 2.1. Location of the Artifact

The triangulation point is in the current territory of Poland, in Upper Silesia, in a place where silver, lead ore, iron ore, and more recently hard coal were mined. The hill, called "Sucha Góra" (German "Trockenberg"), is located on the border of two Silesian cities, Bytom and Tarnowskie Góry.

This granite pillar identifies one of the most important points of the triangulation network on the European continent. It was established during the first world-wide and pan-European measurements of the shape of the Earth, carried out in the first half of the 19th century. These were the times of the constitution and development of modern geographic, cartographic and geodetic research. Triangulaon point "Sucha Góra" is a granite pillar with a square cross-section of 30 × 30 cm, protruding from the ground by several dozen centimeters. It is well oriented; on its northern wall, the letters "TP" are carved, on the southern side there is a symbol of an equilateral triangle. It is a geodetic mark used by the Prussian, German, and then Polish topographic services from the mid-nineteenth century to mark the measurement points of the first-order geodetic networks. (Figure 1).

### 2.2. Working Methods

The research work began with the search for available documents identifying the history of the triangulation point "Sucha Góra-Trockenberg" as a point of the first order triangulation network preserved in cartographic materials and as the starting point of local geodetic networks used in mining until 2000 in the so-called Upper Silesian Coal Basin, located in the territory of Poland and the Czech Republic.

In order to determine the progress of triangulation and the dates the geographic coordinates of the point were determined, field journals were analyzed, as were other materials in the resources of official units (the Department of Geodesy and Cartography and the Department of Monument Protection of the Bytom City Office). An analysis of maps containing information about the existence and importance of point triangulation, developed in the 19th century by both military and civil services, was performed. We reviewed the collections of archival maps from the early 1820s of the development of the industrial area in Silesia, both German and Russian, to prove that the point was the beginning of geodetic networks on the basis of which mines and industrial plants were built in the German and Russian basins, and

we tried to trace its relationship to the Austrian cadastre [9–12]. All sources, cartographic, text, and photo, are characterized in the section "Results". We present the results of our field research in the following order: (a) the history of this point witnessed the development of triangulation and geodesy in Europe, while proving its uniqueness as a link between the three largest triangulation networks on this continent, (b) we show the role this site had in the triangulation-based measurements of the Earth.

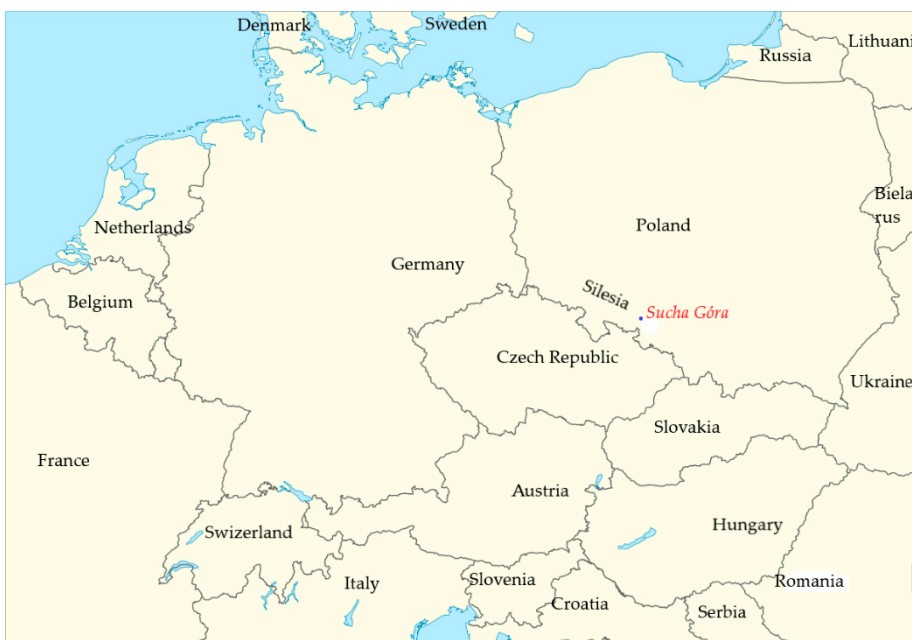

**Figure 1.** Sucha Góra in Europe (made by us, based on Central Europe Location Map Wikimedia Commons).

## 3. Results

### 3.1. Historical Background

Triangulation measurements carried out in the years 1669/1670 by Jean Picard (1620–1682), founder of the French Academy and astronomer of the Paris Observatory, together with the astronomer Adrian Azout (1622–1691), marked the beginning of the 300-year-long epic of modern Earth measurement, conducted both for scientific and economic, as well as political, purposes. These initial measurements established a course of action and a procedure that has become a kind of a compass for surveyors from all over Europe. The determination of the degree of latitude in the Paris meridian became the first realized goal of modern triangulation. Further measurements of the length of this meridian under the direction of César-François Cassini de Thury (Cassini III, 1714–1784) led to the fact that in 1733 this meridian was already designated throughout France, thus creating the first in the world geometric matrix used for creating the "Carte géométrique de France". The triangulation of France was completed in 1744. It soon served as a model for geodetic and cartographic works for other countries. [13].

### 3.2. Müffling Triangulation and "Urmesstischblatter"

Prussia was one of the European countries that undertook large-scale triangulation measurements. The undertaking of geodetic and cartographic works in this country was caused by the urgent need to publish precise military operational maps. In Prussia, no measurements had previously been carried out based on uniform mathematical, geodetic, and topographic foundations. Thus, the measurement methods used up to that point turned out to be insufficient to create sufficiently accurate maps [14]. The government of Prussia ordered organizational changes in geodetic and cartographic services. The domestic survey, which was largely carried out by the army, was transferred, in 1796, to the

General Staff Quartermaster. From 1805, measurements for the civil administration were taken over by the Prussian Statistical Office. In 1810, this office was ordered to prepare new maps with large scales (1:25,000), which were also to be used for operational purposes (after being reduced to 1:86,000 in the west and 1:100,000 in the east) [15]. Looking for a way to increase the accuracy of the measurements, triangulation was used, which was quickly used to make more precise measurements.

At the behest of the King of Prussia, Wilhelm III, Marshal Friedrich Karl Ferdinand Freiherr von Müffling, closely cooperating with General Johann Wilhelm von Krauseneck, took over the management of the triangulation work. In order to standardize the method of conducting geodetic works, Müffling issued the "Instruction für die topographischen Arbeiten des Königlich Preußischen Generalstabes" (1821), which became the official procedure for conducting triangulation works and drawing maps [16,17]. It was based on the work of Carl v. Drecker, who dealt with the standardization of cartographic measurement methods and markings used on maps. As a result of his work, an official textbook of the methodology of cartographic works carried out by the Prussian army after 1816 was printed [17,18].

After the conclusion of the Paris Peace in 1814, the cartographic duties were transferred again to the Geodesy Department of the General Staff (Generalstab, Abteilung Landesaufnahme), commanded by General CV v. Müffling from 1821. In accordance with the orders of the Minister of War HLL v. Boyen, Müffling was to manage the military production of an operational map for all of Germany [19,20]. According to Müffling's instructions, the map was to be drawn at a scale of 1:100,000 (Generalstabskarte). The final decision on mapping, using a scale of 1:25,000, allowed for the collection of accurate data for the whole of Prussia, while at the same time giving the opportunity to generalize the content and create overview maps at smaller scales. The order to cover all of Prussia with a topographic map guaranteed the ruler better control over the new territories, so it was important to create a uniform work that at the beginning would be more accurate than the existing maps [21]. The mathematical guidelines that served as the basis for the map and folio drawing were as follows: 1:25,000 scale, which corresponds to a linear scale of 8 inches per mile with a length of 2000 Brandenburg rods; Multi-plane Müffling projection, folio format 6' wide and 10' long, corresponding to an area of approximately 125 km$^2$ [12,19,22–24]. This was due to the technology of map preparation based on flat tabular measurements using the previously marked triangle mesh (first triangulation).

According to Müffling's instructions, the triangles constituting the basic triangulation network had to have sides from 25 to 50 km long, angles greater than 24° and vertices that are mutually visible. The triangles of the basic network were divided into smaller ones, that is, from 8 to 25 km long, and those into even smaller ones (of the third order), with sides from 1 to 8 km long. Sometimes, fourth-order triangulation on the sides of triangles was introduced in the range from 1 to 3 km. Astronomical measurements at the starting points were carried out with the greatest accuracy using the most precise instruments available, carefully verifying the location of the points. The combination of triangulation with astronomical measurements was aimed at accurately mapping the Earth's surface (assigning latitude and longitude) to the map surface. Based on the above recommendations, in the years 1816–1832, teams of von Müffling and Krauseneck conducted triangulation measurements, which were an extension of the networks previously made in the western parts of Germany (Tranchot works). Starting the triangulation from the Zachs observatory on the Seeberg mountain, the measurement work was carried out near Gotha in the vicinity of Wrocław. On the basis of such a determined triangulation chain, in 1821, Müffling determined the flattening of the Earth ellipsoid at 1:310 [25].

Work on the triangulation of Silesia continued. Measurements were carried out through Strzelin, Annaberg (The Saint Anna's Mountain), as far as Pszów in Upper Silesia. The triangulation measurements probably also covered Sucha Góra, as evidenced by its marking with a triangle symbol on the Urmesstischblatt map No. 3258 (Tarnowitz), mapped in 1827 by lieutenant von Diezielski from the 32nd infantry regiment. This map was made

in accordance with the cartographic guidelines contained in the aforementioned Müffling's instruction from 1821 [26], (Figure 2).

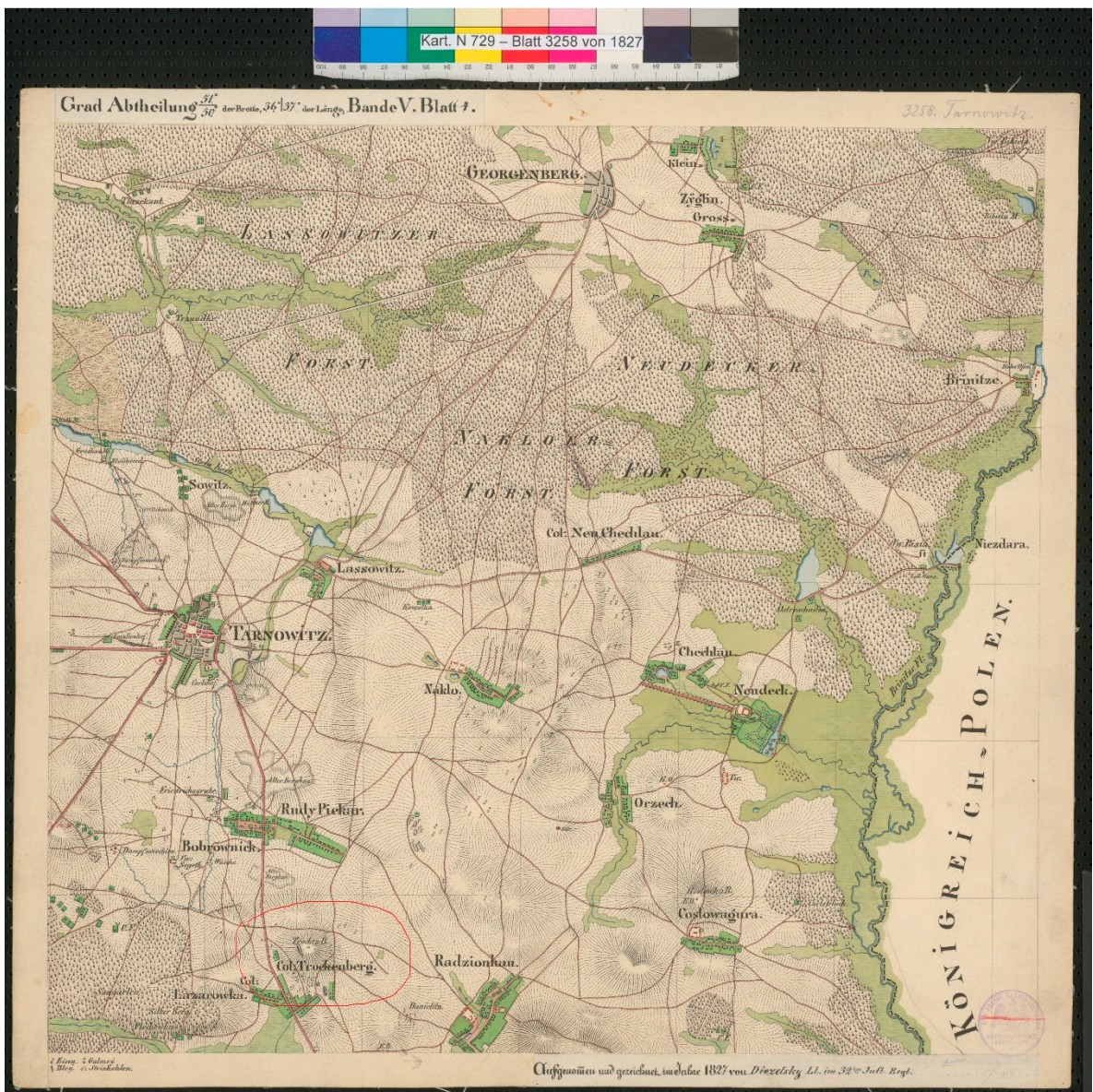

**Figure 2.** Urmesstischblatt map No. 3258 Tarnowitz, 1827; including (Trockenberg (Sucha Góra) is in red circle) [26].

According to this manual, the individual sheets of the Urmesstischblatter map were 10′ in longitude and 6′ in latitude. They were inscribed in a geographical grid composed of parallels every 1 degree and meridians counted from the Ferro meridian. In this way, each of the square cells of the 1° × 1° mesh contained 60 sheets (6 columns (sheets) × 10 strips (lines)). The description of each sheet of the map contained the data of the mesh of the grades (Grad Abtheilung) and the number of the band (Bande) expressed in Roman numerals (IX) counted from south to north and the number of the next sheet (Blatt) in the given strip expressed in Arabic numerals (1–6).

The mentioned sheet, 3258 (four-digit numbering was given to the sheets later and is marked in pencil in the upper right corner of the map), is included in the mesh at 51°/50° latitude and 36°/37° east longitude with respect to the Ferro Meridian. It belongs to lane No. V (frame latitude from 50°24′ to 50°30′) and sheet No. 4 (longitude between 36°30′ and 36°40′ east of Ferro, i.e., 18°50′ E and 19°50′ E in relation to Greenwich). The content

of the maps is rich, although it does not yet contain a contour drawing. Their detailed analysis is provided by the studies by [12,24].

The linear interpolation carried out by the authors of this work allows for the determination of the Trockenberg topographic point from the Urmesstischblatter map at 50°24′40.55″ N and 36° 32′37.12″ E in relation to Ferro, i.e., 18°52′37.12″ E relative to Greenwich. For Silesia, in the years 1824–1828, 134 maps were drawn with a scale of 1:25,000, based on plane table measurements made after the completion of the first triangulation [23,24].

*3.3. Sucha Góra, an Astronomical Point and a Place Where the Prussian and Russian Triangulation Networks Were Connected.*

In 1843, the Russian lieutenant-general Shubert informed General von Krauseneck, the then chief of the general staff of the Prussian army, about Russia's intention to triangulate the Kingdom of Poland and at the same time expressed a desire to connect it with the Prussian triangulation network [27]. Referring to this proposal, general von Krauseneck suggested areas near Toruń on the Vistula and near Grodziec in Upper Silesia as suitable for such a connection. In Silesia, the connection to the Prussian triangulation network was to take place through the points of Góra Św. Anna and Pszów. In the following years, the case was transferred to the head of the Triangulation of the Kingdom of Poland, Lieutenant General Tenner, and to the head of the Trigonometric Division of the Prussian General Staff, Colonel Baeyer. In 1845, Tenner began triangulation measurements in the Kingdom of Poland. Overall, three baselines were measured, one near Warsaw, the second near Tarnogród on the border with Galicia, and the third in the area of Częstochowa city near the border with Silesia. The main point of the network was the Warsaw observatory. Triangulation in Poland was the best work done by Tenner. The triangulation of the Kingdom of Poland included 219 measurement points of the 1st order and 2112 points of the 2nd and 3rd orders. During this work, Tenner wrote a detailed instruction on how to perform triangulation works. [28].

It is worth mentioning that in Tenner's astronomical and geodetic works, Adam Prażmowski, one of the most outstanding European astronomers of the 19th century, contributed a large share by, in August and September 1849, measuring the latitude of the Markowice point (50°33′41.607″) using astrometric methods. This point connects directly to the Sucha Góra point in a triangulation network [29]. In 1850, Tenner met Baeyer, during which the course of further cooperation related to the merger of Prussian and Russian triangulation was established in detail. Particular attention was paid to recording the measurement points in the field. While the Russian side had already attached great importance to the stabilization and fixation of measurement points with durable structures made of rock and ceramic materials, in the case of the previous practice on the Prussian side, markings made of perishable material (wood) played the role of base points.

Thanks to this agreement, between 1850 and 1852, an astronomical and geodetic point was recorded on the Sucha Góra. A detailed description of the construction works related to the preparation of the measuring point can be found in the Baeyer report (Die Verbindungen den Preussischen und Russischen Dreiecksketten bei Thorn und Tarnowitz. Ausgeführt von der trigonometrischen Abtheilung des Generalstabes) [30]: The measuring instrument was placed on a cast iron plinth made in the Royal Iron Foundry in Gliwice. A cast iron pipe, four feet and nine Prussian inches long and one Prussian foot diameter, was cut off at the ends perpendicular to the central axis. A 3/4-inch thick, 2-foot diameter circular iron plate was attached to each end of the pipe, the center line of the pipe being defined by fine tapered holes. A 3/4-inch diameter recess was drilled around the center of the base which (to define the center of the point) was filled with lead. The base plate had four symmetrically spaced holes through which the anchors passed, provided with threads and nuts for attaching the plate to it. To stabilize the point, a 6-square-foot, 5-foot-deep pit was dug. The hole was originally intended to be 6 feet deep, but hard limestone rock prevented this. A 3 1/4-foot high foundation was made in the trench, to which the base plate was bolted, along with the pipe and the measuring tabletop. The floor beams were placed

above the excavation so that the floor was completely insulated from the iron pillar of the measuring instrument [30] (Figure 3).

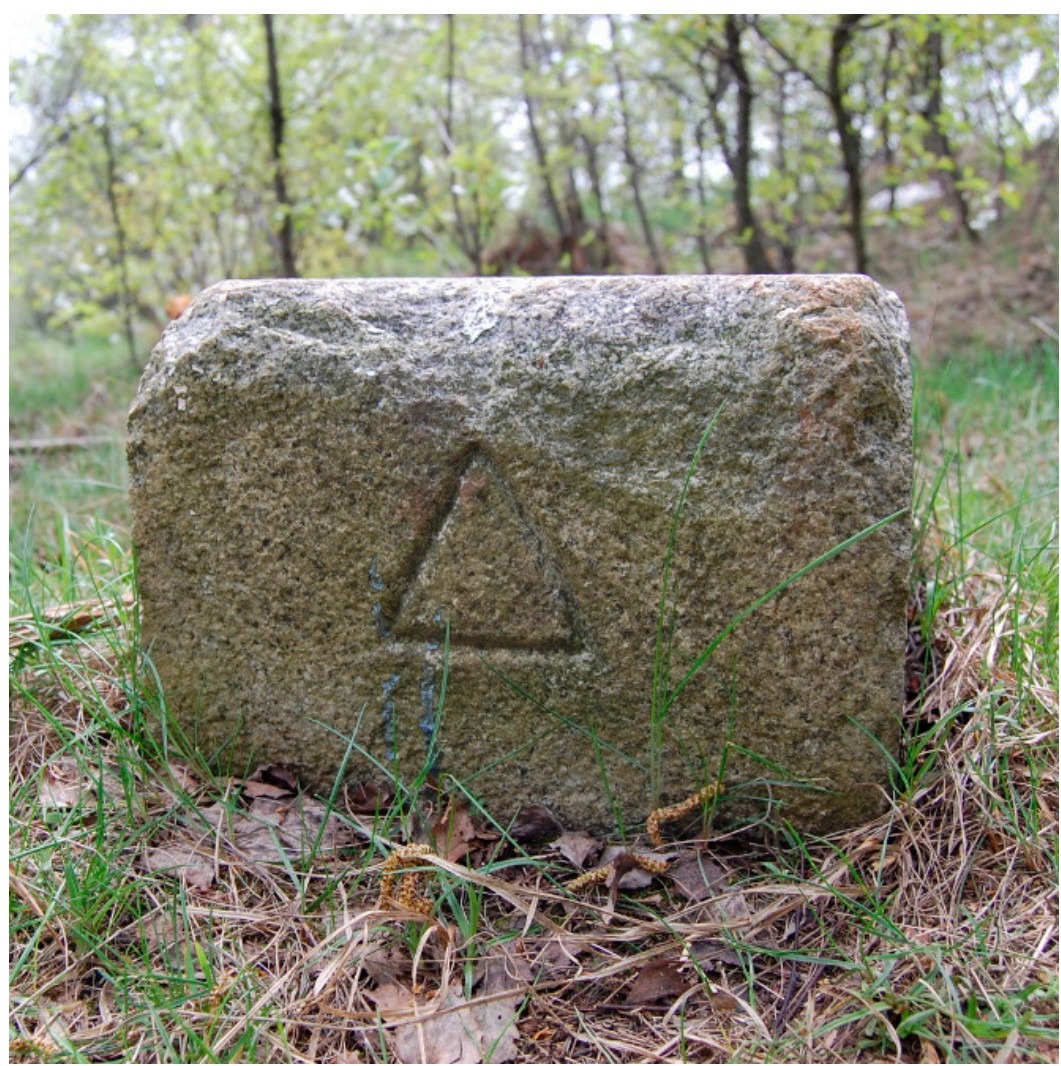

**Figure 3.** The astrogeodetic mark "Sucha Góra".

As a result of the astronomical and geodetic works carried out by Baeyer between 17 and 29 June 1852, examining the position of alpha stars from the constellation Taurus (α-Tau) and Leo (α-Leo) and the Pole Star (α-UMa), the latitude of the Sucha Góra point was determined at 50°24′44, ″01. [30]. Network-binding triangulation works on the Prussian side were carried out in 1852 and 1854 [30]. Triangulation works on the Russian side were carried out in 1854 [29]. In addition, the height of the Sucha Góra point above sea level was determined. The measurement was made in relation to Góra Świetej Anny (Mount St. Anna, German: Annaberg) (for which the absolute height had previously been determined). The height of the cast iron plate above the Baltic sea level was determined at 180.0809 toise above sea level [30].

*3.4. Fixing Sucha Góra Triangulation Point during the Measurements of the Arc of 52° Parallel of North Latitude (Längengradmessung in 52° Breite und Mitteleuropäischen Gradmessung)*

In 1857, Baeyer published the results of his work on the connection of the Prussian and Russian triangulation networks in the vicinity of Toruń and Sucha Góra [31]. The two astronomical authorities, Professor Johann Franz Encke (1791–1865), director of the Berlin observatory, and Friedrich Georg Wilhelm Struve, founder and first director of the Pulkovo

observatory, who acted as reviewers of this work, stated that the results were of very high quality. Struve's detailed review was published as an appendix to Baeyer's work [31]. It is the first documented source where the longitude of the Sucha Góra point appears, which is, according to Struve's calculations, 16°32′35.00″ for Baeyer's measurements and 16°32′31.67″ for Tenner's measurements (the lengths relative to the Paris Observatory).

He drew attention to the high concordance of the results made independently by the Russians and the Germans, but also to the differences in the determination of the latitude and longitude of Laplace points, reaching 3″ in longitude and latitude. He suggested that observers should make consistent astronomical measurements using the same equipment with the highest possible accuracy, which would allow determination of local vertical deviations. He also recommended the use of a recently invented electric telegraph to measure differences in longitude.

In summing up his work, Struve pointed out that there was already a continuous European triangulation network between Hammerfest in the north, Cap Marco in the south, Valencia in the west and Astrakhan in the east [31]. This prompted Struve to undertake an ambitious project to measure the length of the arc of the 52°N latitude parallel and its point of connection Trockenberg-Sucha Góra. This project significantly influenced another, equally important from the point of view of the history of geodesy projects measuring the European meridian [32], about which we will write later in the article. Figure 4 shows a schematic course of the triangulation network in Western and Eastern Europe related to the measurement of the 52° north latitude arc and its connection point: Trockenberg i.e., Sucha Góra.

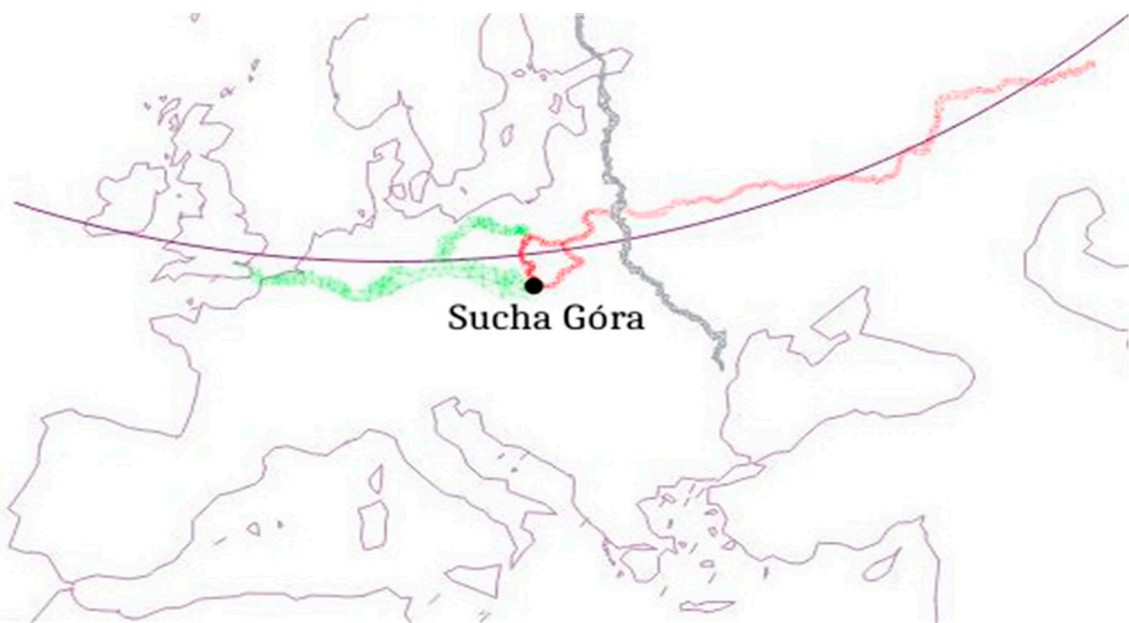

**Figure 4.** Schematic course of the triangulation network in Western (green lines) and Eastern (red lines) Europe. (Map made by us based on [27]).

### 3.5. Measurements of the Central European Meridian (Mitteleuropäischen Gradmessung)

In 1861, Johann Jacob Baeyer proposed his famous project to measure the meridian arc in Europe [33]. The most important innovation in relation to the existing geodetic projects was the use of a triangulation network located throughout the measurement area instead of the previously used lines of triangles along parallels and meridians. This plan was met with great interest both by the Prussian government and abroad. This led to the creation of the Mitteleuropäische Gradmessung (Association pour la mesure des degres dans I'Europe centrale)-soon renamed Europäische Gradmessung (Association pour la mesure des degres en Europe), then the Internationale Erdmessung (Association geodesique internationale) and finally taken over by the Association Internationale de Geodesie [32].

The diagram of the triangulation network for the measurements of the Central European Meridian is shown in Figure 5.

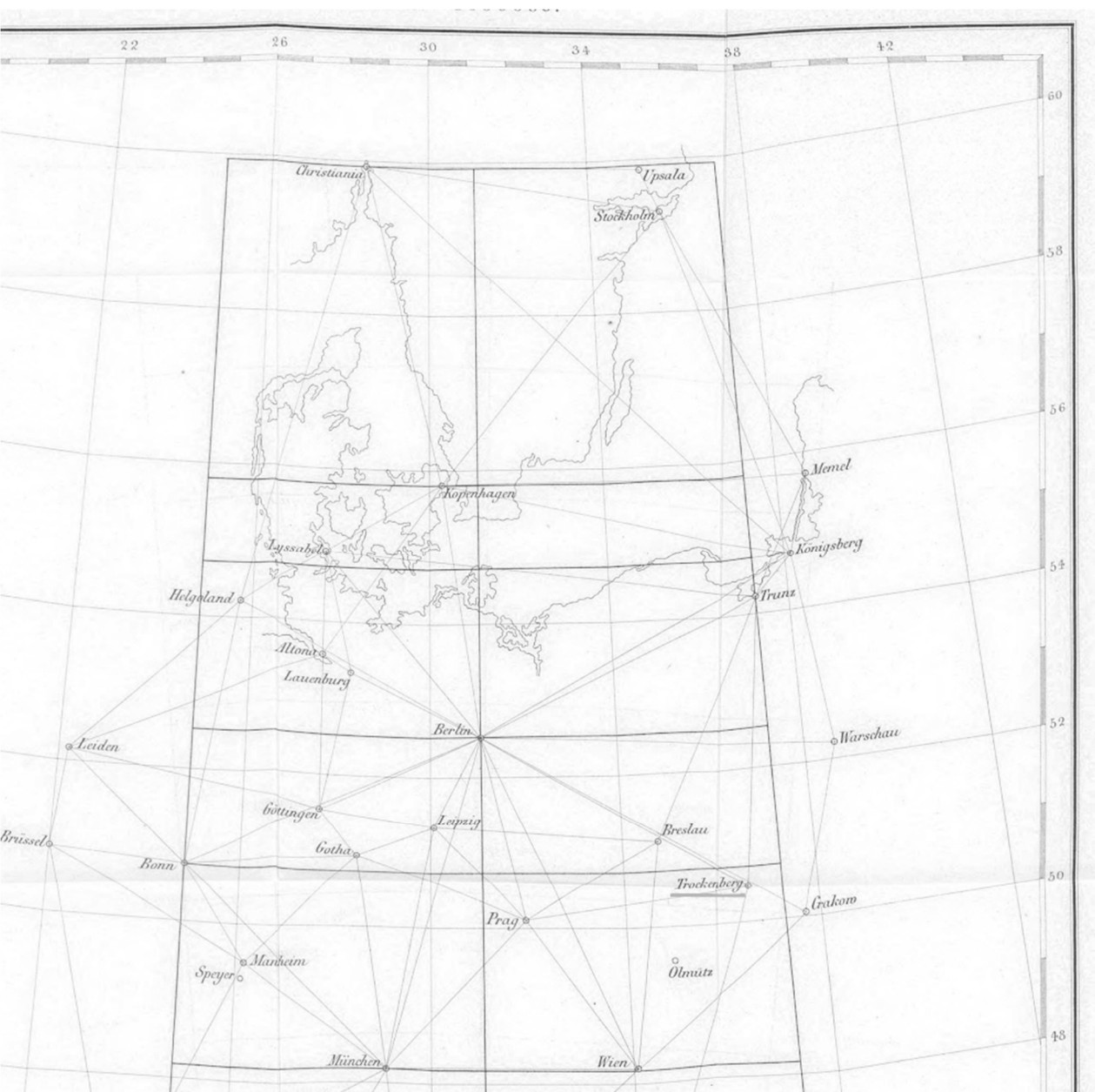

**Figure 5.** Fragment of a triangulation network for measurements of the Central European Meridian [33]–Trockenberg (Sucha Góra) is among 38 selected points of the measurement network.

In the list of measurement points, Baeyer [34] gives the geographical coordinates of the point "Trockenberg" 50°24′44″ N in latitude and 36°32′35″ E in longitude (taking Ferro as the prime meridian). The longitude of Trockenberg was re-measured using a more accurate measuring method using an electric telegraph in 1888 by the team of Dr. Th. Albrecht, which included the observers E. Borrass and H. Richter from the "Königlichen Geodätischen Instituts" [35].

The longitude of the Sucha Góra point was then established as 18°52′37.01″ east of Greenwich, which, taking into account the difference between the geoid and the reference ellipsoid (Bessel), took the value of 18°52′39.9732″ and was used as such in cartographic systems until the middle of the 20th century.

*3.6. Sucha Góra Triangulation Point in the Twentieth Century and in Recent Times.*

The turbulent history of the Upper Silesia region in the first half of the 20th century resulted in the transition of the Sucha Góra point to the Polish administration. This point was included in the triangulation network of the Polish State, which included points from three different triangulation systems (German, Austro-Hungarian, and Russian). Points close to the borders of the former partitions played a key role in connecting and aligning the new triangulation network. One of such points turned out to be the Suchogórski point, precisely measured in the Baeyer and Tenner campaign in 1848–1852. It is worth mentioning here that from the end of the 19th century the point of Sucha Góra served as the mapping center (point (0,0)) of all mining maps of the Upper Silesian Coal Basin. After the Second World War, the Sucha Góra point retained its role as the zero point for mining maps. The solid stabilization of the point meant that in the 1980s it was used as a measurement base point for checking geodetic devices used in mining (gyrotheodolites) [36]. Only changes related to the introduction of satellite measurement techniques at the end of the 20th century and the slow decay of mining exploitation in Upper Silesia resulted in the Sucha Góra point gradually losing its importance. A new PL2000 state coordinate system was introduced, based on the Gauss Kruger projection, which replaced the Sucha Góra system on mining maps. Currently, the Sucha Góra point is a point of the horizontal base control with the identification number 521401800. It is also a so-called perpetual point.

## 4. Discussion

The Sucha Góra point carried and still carries many values, meanings and information, which we have scored below: it is one of those artifacts present in the landscape that has been a witness of and a monument to the development of science and technology. It has documented the history of the development of modern methods for measuring the shape of the earth and is one of the most important elements of the system in which it was made. Due to the times in which these measurements were made, it has documented geopolitical changes in Europe and in the world. Due to the place where it was located, it has documented and has been an element of the landscape changes that have taken place in connection with the construction of a large, industrial basin in this part of Europe. The scientific and cultural heritage of the "Sucha Góra" point is valuable, but the question remains how to popularize this point and introduce it as one of the great monuments of geodesy, geography, and geopolitics.

So far, the most well-known objects in this area have become the Struve's meridian, observatories in Greenwich, Pulkovo, Ferro, and Gusterberg in the Czech Republic, and the equator monument in Ecuador. There are also a number of smaller objects, such as the Normaal Amsterdam Peil, Brušperk, the observatory in Warsaw, Vienna, and Leipzig. Time will show how the "0" point in Sucha Góra will be treated, and what its role will be in the local and European heritage. The story of the point combines several threads. It also awakens specific reflections that will be different for the European reader and for the Polish one.

Earth measurements carried out in the 19th and 20th centuries accompanied political changes in Europe and the world. This was documented in an official way, using cartographic and geodetic methods. The purpose of these measurements was to determine the size of the domains and to facilitate control over the land owned by them. The leaders of the measurements were: Great Britain, France, Germany, and Russia. In Central Europe, the Russians led them through the areas of Poland, a country that had lost its independence as a result of geopolitical change in the 18th and 19th centuries. The Kingdom of Poland, dependent on Russia, became the area of connection of two large measurement networks. It should be emphasized that both of the Tenner and Bayer geodetic surveys were carried out in the territory of Poland and not on the borders of the German or Russian empires.

The interesting location of this point in the vicinity of a UNESCO site, as well as its location in the forest and park complex justify undertaking activities related to the conservation and promotion of this cultural heritage site. In cooperation with the local community, local au-

thorities, and lovers of the history of science and technology, the authors of this article express the hope that the Suchogórski point will regain its historical significance in the near future-this time as a monument to European geodesy. The first attempts to obtain financing from the city's civic budget have already been made. This has resulted in an increase in the awareness of the local residents about the place and role of the Suchogórski point. In the long-term, we expect to place an observation tower in the shape of a 19th-century triangulation tower above the Suchogórski point, and to create a scientific and cultural theme park nearby—devoted to measuring the Earth. The Suchogórski point itself, due to its stable anchorage in solid limestone rock, can also continue to be used for scientific research as a reference point for measuring vertical displacements (subsidence) of the nearby mining area of the city of Bytom, proving the almost 200-year history of this place.

## 5. Conclusions

We aimed to show the history of the "Sucha Góra" point and document its importance for modern geodesy and cartography's scientific and cultural heritage. We did this by collecting the necessary amount of information about this part of Europe's complex history. Below we have presented the implications that result from the rich history of Upper Silesia and which are of fundamental importance in the interpretation of the heritage of point Sucha Góra.

Measurements and calculation of Earth's shape are fundamental scientific issues that were solved in the 19th century. The ideas of joint, international geodetic works with the participation of scientists of Polish nationality laid the foundations for the International Union of Geodesy. They perhaps even became part of the processes which eventually led to the emergence of the Europe of Nations. During the Earth's measurements, in the 19th and 20th centuries, several important processes were taking place in Upper Silesia that determined this region's identity. One of them was industrialization. The contemporary Silesian population, which determined their national identity in industrialization times, may include the "Sucha Góra" point in the cultural heritage. The "0" point is inscribed in the development and shaping of the working class in Silesia [37,38]. Thus, geopolitical determinants will play a vital role in shaping the heritage of Sucha Góra, which today are still emphasized in the discourse on the shape of Europe and the borders of Poland and Silesia.

**Author Contributions:** Conceptualization and methodology, M.L., M.D.; validation, M.L.; writing—original draft preparation, M.L, M.D.; writing, review and editing, M.D., M.L.; visualization, M.D.; funding acquisition, M.L. All authors have read and agreed to the published version of the manuscript.

**Funding:** This research received no external funding.

**Acknowledgments:** Authors would like to express their kind acknowledgements to Tomasz Białożyt, Henryk Lamparski, Mariusz Meus (Honorowy Południk Krakowski) and Tomasz Nycz (GiS w Górach) for many hours of lively discussions about Sucha Góra and historical European triangulation networks.

**Conflicts of Interest:** The authors declare no conflict of interest. The funders had no role in the design of the study; in the collection, analyses, or interpretation of data; in the writing of the manuscript, or in the decision to publish the results.

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
