# Peer review of "“Sucha Góra” (“Trockenberg”)—The Triangulation Point Doomed to Be Forgotten?"

_land, doi:10.3390/land10020109_

Round 1

Reviewer 1 Report

I read through the article titled " 'Sucha Góra' ('Trockenberg') – The Triangulation Point doomed to be forgotten? " and found the content to be of interest to the readers of this journal. The authors present an interesting story of the Sucha Góra triangulation point's origins and historical development. I was very interested to read about the history or this triangulation point and how the authors wove together the geodetic and political importance of this triangulation point. I offer three main points of feedback on ways to improve the overall quality and readability of the article. 

First, in specific places the English writing is difficult to follow. Some of the sentences are not complete, the phrasing is awkward, and the flow is not very smooth. I made attempts to edit what I could, but without knowing the full scope of the authors' thoughts and facts, I could only provide superficial edits. A good proof-reading and editing would help present a more coherent, well-organized, and, readable piece of research. 

Second, I think the organization of the article could be improved. Based on my read of the article, I see two ideas that are being expressed here. First, there is the history of how and why the Sucha Góra triangulation point was established and the importance it played in the geodetic network of the area. With the exception of some of the writing, this discussion was more or less ok. Second, there is the broader importance that the Sucha Góra triangulation point played in shaping the politics and culture of the region. This discussion is not well presented and is at times rather difficult to follow. I recommend re-casting the organization of the article in this fashion: there should be two main sections. One of the sections presents a history of the surveying and establishing the Sucha Góra triangulation point. The other section would discuss the implications of establishing the Sucha Góra triangulation point and how this establishment impacted the politics and culture of the region. As I point out on page 10, the authors do a fine job of highlighting five ideas as to how the Sucha Góra triangulation point impacted politics and culture of the region. I encourage the authors to use these five points as a way to frame what I am referring to as the second main section of the article. Then, use these five discussion points as ways to organize and report on the political and social impacts of the Sucha Góra triangulation point.

Third, I would remove references or statements to the effect of the "great colonial powers" and their supposed triumphs. In this day and age, there is general agreement that the colonial powers were not great and did much harm to those they sought to colonize. Instead, I would cast the discussion away from the idea of colonial powers altogether and instead frame and focus the writing more as a tribute to the individuals and how they implemented the technological achievements in geodesy and mapping that was needed to survey and establish the Sucha Góra triangulation point. 

I hope you will find these comments helpful in your revision. I have also uploaded a PDF of your article complete with my comments. 

Author Response

Dear Reviewer

Thank you for your in-depth review and the many valuable comments that we used to improve the text. We tried to respond to each remark. We hope we understood their suggestions and questions and have attached a PDF of your comments along with our responses. The frequently used word "revised" means that your corrections have been made in the text sent to the editor. In case of questions - we tried to answer in the comments of your pdf. As a result of your review, we have changed large fragments, especially those concerning colonialism. We removed and corrected controversial sections. As Poles who know the history of their own country, we cannot praise colonialism! Unfortunately, perhaps due to language problems, you got this impression.

A native speaker has improved the text sent back to the editor. We hope this makes the text better.

Thank you for your suggestion to edit the entire article. However, we now want to stay with the original version—the comment is in the PDF.

However, we have edited the Conclusions and Discussion. We have included a justification for why we did not change the article, as you suggested. Additionally, we would like to inform you that we want to write one more article to refer to these scenarios. It will concern the analysis of mining and geological maps created based on the astrogeodetic point, Sucha Góra.

We hope you enjoyed the answer.

We enclose our regards.

Reviewer 2 Report

Dear Authors,

The article is interesting and accessible for the potential reader. In overall manuscript is nicely written and presented data are important for the national and international community. There are some issues of medium importance that I would like to address in the attached file. 

Author Response

Dear Reviewer
Thank you for your in-depth review and valuable comments that we used to improve the text. We tried to respond to each remark. We hope we understood their suggestions and questions. As a result of your review, we have changed large fragments.
A native speaker has improved the text sent back to the editor. We hope this makes the text better.
In the text below, we would like to comment on the remarks.
"INTRODUCTION General comments: In my opinion the second paragraph should be incorporated in the MATERIALS AND METHODS section as it in fiact descibes the object of the study. Moreover the could be few more citations in this paragraph as these informations were not discovered by the Authors. First paragraph require adding of some citations for the same reason.
We have kept the original layout of the text. In this way, we wanted to highlight the location of the Sucha Góra point in Silesia and Poland's space. We introduced few citations because we relied solely on maps and historical texts in the course of our research. We wrote about it in the "Materials and Methods" section.
MATERIALS AND METHODS General comments: Please list all the sources used for the study in this section or clearly explain that all of them are listed or mentioned in the DISCUSSION section (I believe they are). I doubt that such an detailed technical description on the Müffling's instructions (lines 192-204) is necessary for the topic of the paper.
We listed all sources in the "Results" and "Discussion" sections. We have kept the information regarding Muffling's instructions. According to these instructions, the godets created the point in this place for the first time in its history.

DISCUSSION General comments: Summarizing of the article should be included on the CONCLUSIONS section. I suggest to rewrite the first sentence or even delete it.
We have redrafted the "Discussion" and "Conclusion" sections in line with this suggestion. In the revised version of the article, the summary remarks are now included in the conclusion.
In all parts of this article, we have implemented corrections contained in detailed comments.

We hope you enjoyed the answer.
We enclose our regards.
